# Autonomous Airline Revenue Management: A Deep Reinforcement Learning Approach to Seat Inventory Control and Overbooking

**Syed Arbab Mohd Shihab** [1]  **Caleb Logemann** [2]  **Deepak-George Thomas** [3]  **Peng Wei** [1]

## Abstract

A revenue management system is a real-time decision making system that maximizes earnings by controlling inventory and pricing according to consumer behavior prediction on micro-market levels. It is the key piece to generate profits in airline, rail, cruise, hotel and rental car industries. In this paper, the seat inventory control and overbooking problem of airline revenue management has been formulated as a Markov Decision Process and then solved by using Deep Reinforcement Learning to find the optimal policy, one that maximizes the revenue for each flight. Multiple fare classes, concurrent continuous arrival of passengers of different fare classes, overbooking and random cancellations that are independent of class have been considered in the model. To generate data for training the agent, a basic air-travel market simulator was developed. The performance of the agent in different simulated market scenarios was compared against theoretically optimal solutions and was found to be nearly close to the expected optimal revenue. This work is among the first few ones to address real-world airline revenue management by using deep reinforcement learning and it has been invited by American Airlines and Airline Group of the International Federation of Operational Research Societies (AGIFORS) for seminar talks and potential collaborations.

[1]Department of Aerospace Engineering, Iowa State University, Ames, Iowa, USA [2]Department of Mathematics, Iowa State University, Ames, Iowa, USA [3]Department of Mechanical Engineering, Iowa State University, Ames, Iowa, USA. Correspondence to: Syed A.M. Shihab <shihab@iastate.edu>.

*Reinforcement Learning for Real Life (RL4RealLife) Workshop in the $36^{th}$ International Conference on Machine Learning*, Long Beach, California, USA, 2019. Copyright 2019 by the author(s).

## 1. Introduction

### 1.1. Motivation

Few markets are as fiercely competitive as the current air travel market. This heightened competition dates back to the deregulation of the airline industry in 1978, which allowed US airlines to freely set up their route network and quote fares for their itineraries. Since then, airline corporations have been relying on *revenue management* (RM) system, a decision support tool designed for maximizing the total expected profits generated from all their flights (Belobaba et al., 2015). RM systems use differential pricing to determine a range of fare classes and their fare levels to exploit the differences in willingness to pay (WTP) of passengers in any given Origin-Destination (O-D) market. A combination of varied restrictions and service amenities is used to create separate fare classes. Then, for a given set of fare classes, aircraft capacity and schedule, RM systems use yield management or seat inventory control for allocating seats to each of the fare classes to protect seats of higher fare class passengers from lower ones. Also, in order to prevent losses in revenue due to certain customers cancelling their tickets or not showing up, airline corporations overbook their seats. But, if the overbooking process is not done optimally, it leads to situations where the number of passengers showing up for the flight is more than the seats in that fare class. Subsequently, a few passengers have to be denied boarding. This leads to airlines facing losses in at least one of the two ways. Firstly, the displaced passenger(s) have to be compensated for their distress in the form of expensive vouchers. Secondly, if the passenger is not adequately compensated, a goodwill cost is incurred (Gosavi et al., 2002). These two costs combined is referred to as the *bumping cost* in this paper.

The combined problem of seat inventory control and overbooking has been examined in this paper. Conventional seat inventory control techniques and overbooking models are strongly affected by the accuracy of the demand forecasting process and modeling approach. Modeling the problem as an MDP and using Reinforcement Learning (RL) can overcome the limitations of conventional yield management techniques as this approach, in theory, does not require formulating complex mathematical models for

revenue computation and passenger demand forecasting. In real life, however, some demand forecasting will still be necessary while training the RL agent on historical booking data to capture the change in demand of passengers resulting from the agent taking exploratory actions which differ from that taken by the airline in the past. Additionally, RL is well suited to perform well under uncertainty. The airline RM problem in real world has a large state space. This paper overcomes this challenge by implementing a Deep Q-learning Network which can approximate the state-action values by interacting with the air travel market simulator.

### 1.2. Related Work

Howard addressed the overbooking problem assuming the airline did not divide the cabin into different fare classes. The problem was modelled as a Markov Decision Process (MDPs) and the optimal policy was found using the value iteration algorithm. However, the computational limitations of the value iteration algorithm made this technique unfeasible to implement on large-scale problems (Howard, 1960). Brumelle et al. tackled the seat allocation problem for several fare classes using the Expected Marginal Seat Revenue (EMSR) technique, a popular model used by the airline industry. The problem was formulated on the assumption that ticket requests for high fare classes are placed after the requests for lower fare classes have been made (Brumelle & McGill, 1993). Lee et al.(Lee & Hersh, 1993) explored the seat inventory control pertaining to airlines. The problem of optimally deciding on booking requests for a booking class at a specific time was investigated. Discrete-time dynamic programming was implemented to arrive at an optimal policy. Such an approach becomes impractical when the number of states increases or if time is treated as a continuous state variable in the problem. Subramanian et al. also addressed the seat allocation problem for several fare classes while taking into consideration the possibility of overbooking, cancellations and absentees on the day of the flight. The problem was modeled as a discrete time Markov Decision Process and an exact solution was found using dynamic programming through backward induction. The algorithm was implemented on a real-life airline dataset confirming its computational feasibility. However, their model was based on the assumption that probability of cancellations was not dependent on fare-classes (Subramanian et al., 1999). Gosavi et al. formulated a similar problem with two major differences. They did not assume that cancellation was independent of fare classes. Additionally, the problem was modeled as a Semi Markov Decision Process (SMDP) instead of MDP. They developed a novel algorithm $\lambda$-SMART to solve the SMDP. The algorithm was compared against EMSR and it was found to outperform EMSR (Gosavi et al., 2002).

The remainder of our paper is organized as follows. The theoretical basis of our work has been described in the background portion. Thereafter, the problem description section elaborates on the MDP formulation and the simulator used to generate our data. Subsequently, the techniques used to solve the MDP and the outcomes are given in the solution and results sections.

## 2. Background

### 2.1. Markov Decision Process

A Markov Decision Process (Bellman, 1957) is generally composed of four components: a set of all the states $s$ referred to as state space $S$, $(s \in S)$, a set of all actions a given by the action space $A$, $(a \in A)$, a reward function $R(s, a)$ which depends on the current state and action, and a transition function $T(s, a, s')$ describing the probability that action $a$ in state $s$ will lead to state $s'$. At time t, the agent chooses a specific action depending upon the current state, following the Markovian assumption. Subsequently, the agent probabilistically progresses into a new state according to the action taken and the present state which results in the agent receiving a reward $r$. A discount factor $\gamma$ is generally included in this process that so that immediate rewards are valued more than rewards that could be obtained in the future. It also prevents the sum of rewards from becoming infinite. The MDP is solved to find an optimal policy $\pi$, which deterministically maps the state to an action.

$$\pi : s \mapsto A \qquad (1)$$

Therefore, the optimal policy $\pi^*$ for a MDP can be defined as one that leads to the attainment of maximum cumulative expected rewards (Kochenderfer, 2015).

$$\pi^* = \arg\max_\pi \mathbb{E}\left[\sum_{t=0}^{T} R(s_t, a_t)|\pi\right] \qquad (2)$$

### 2.2. Q-learning

Q-Learning (Watkins & Dayan, 1992) is a popular technique to determine the value of performing an action while in a specific state. The algorithm iteratively returns Q-values by implementing incremental estimation in the direction of the observed reward and estimating future rewards from the subsequent state s'. In order to ensure that the model converges to the optimal value, some amount of exploration is required depending upon the known information of the environment (Kochenderfer, 2015). The optimal action at each state is the one that maximizes the state-action value.

$$Q(s, a) \leftarrow Q(s, a) + \alpha(r + \gamma \max_{a'} Q(s', a') - Q(s, a)) \quad (3)$$

## 2.3. Neural Networks

Many real-world problems have a large state space, where it is impossible to record values for every state and action pair. Furthermore, the agent would not be able to visit all states. So, state-action values that have not been encountered needs to be generalized. This can be done using neural networks, which contains neurons, also known as perceptrons, to approximate the state-action values (Mnih et al., 2015). A perceptron consists of three components: input nodes $x_{1:n}$, weights $\theta_{1:n}$ and a output node $q$. Combining the idea of approximating state-action values using perceptrons and training the agent with Q-learning resulted in a popular approximation method known as perceptron Q-learning.

An inherent drawback of perceptron is that it can model only linear functions. However, a set of perceptrons can be combined to form a neural network which can approximate nonlinear functions. Non-linearity is introduced using activation functions. Sigmoid, Tanh and ReLU are commonly used activation functions. A neural network possesses an input and an output layer with hidden layers between them. The backpropagation algorithm is usually used with neural networks for mitigating the loss function, given by the temporal difference error, to learn the appropriate features and weight (Kochenderfer, 2015). According to the universal function approximation theorem, a feed-forward neural network with one hidden layer, given sufficient neurons and mild assumptions on the activation function, can approximate any real continuous function. Cybenko was one of the pioneers in proving this theorem for sigmoid activation functions (Cybenko, 1989).

## 2.4. Deep Q-Learning

Like perceptron Q-learning, deep Q-Learning also combines the idea of using an approximator and Q-learning. But, instead of using a perceptron, a deep neural network is used. Equivalent to a multilayer perceptron, the Deep Q-learning Neural Network (DQN) has several hidden layers, resulting in a large number of weights as its parameters. Q-learning with backpropagation is used to update the parameters of the neural network such that the loss function is minimized (Hausknecht & Stone, 2015). Since the generation of the succeeding Q-values and the updating of the present Q-values is done by the weights of the same network, other Q-values estimates in the state-action space can also get erroneously updated (Tsitsiklis & Van Roy, 1997). Deep Q-Learning mitigates this issue by employing the following approaches. Firstly, the set of experiences are stored and during training they are sampled uniformly, such that the neural network is always fed uncorrelated data to avoid having bias to more recent data. Secondly, the primary network is updated by a different network, preventing the performance issues that arise when the generating and updating

is done by a single network. Lastly, every parameter is provided with a robust learning rate, alpha, which is updated after taking into account its preceding values (Hausknecht & Stone, 2015).

# 3. Problem Description

## 3.1. Problem Statement

For every flight, the optimal seat allocation and overbooking limits for each fare class needs to be determined such that revenue is maximized. Uncertainty in customer booking request arrivals of each fare class in each flight makes this problem a stochastic one. Moreover, customers typically request bookings at different times prior to any given flight departure. For each booking request, the airline can either accept or deny it. So, a series of actions need to be taken at different points in time till the date of departure, which makes this problem a sequential decision making one. Taking these facts into account, the seat inventory control and overbooking problem has been modeled as a MDP, where the agent does not know the transition and reward models. To find the optimal policy, the agent needs to learn through experience represented by state transitions and received rewards. The data to generate this experience is obtained using an air travel market simulator.

## 3.2. Air Travel Market Simulator

In order to train the agent, an environment was created to simulate the arrival of passengers of different classes wishing to book tickets for the flight. Customers are allowed to reserve seats 1000 days prior to the flight departure. Each class of passengers was simulated as an independent Poisson process. Each test case can specify the expected number of passengers to arrive for a given class. In order to simulate their arrival an exponential distribution is sampled whose mean is the ratio of total time to expected number of passengers. Sampling the exponential distribution gives a list of inter-arrival times, which can them be assembled into a list of timestamps at which passengers arrive. This process results in the number of passengers from each class being distributed according to a Poisson distribution. If a passenger arrives, then the cancellation probability will randomly set whether or not the passenger will cancel at a later time. The time at which the passenger cancels is uniformly distributed along the remaining time before the flight. Therefore, each episode or flight will consist of a list of potential passengers, their class, their booking time, if they will cancel, and if so at what time they will cancel.

Given this data the optimal reward possible can be computed. The optimal policy will be to accept all of the passengers from the highest fare class, and then the lower fare classes in descending order until the capacity is filled or all of

the passengers have been accepted. The optimal reward is then just the fares applied to these passengers. The agent cannot achieve the optimal reward as it requires knowledge of future cancellations and future arrivals, however this can be a useful metric to gauge how well the agent is performing.

### 3.3. MDP Formulation

#### 3.3.1. STATE SPACE

The state space ($S$) vector contains the information generated during the booking process regarding the airline seats. It includes the travel class of the latest customer ($T$), seats that have been sold for the nth class ($b_n$) and the time remaining for the ticket booking process to end ($t$).

$$S = (T, b_1, b_2, ..., b_n, t) \qquad (4)$$

A typical state can be illustrated by the following example. A customer requests a middle class seat 40 days prior to the departure. Additionally, the inventory shows that the number of seats booked in the high, middle, and low fare class are 2, 20 and 20 respectively. In this case, the state space can be given by (2, 2, 20, 20, and 40). The state variable t is continuous while the rest is discrete.

#### 3.3.2. ACTION SPACE

At every time step, exactly one of the two decisions, accept ($a_{+1}$) or deny ($a_{-1}$) can be made. The action space (A) is given by:

$$A = \{a_{+1}, a_{-1}\} \qquad (5)$$

#### 3.3.3. MODEL DYNAMICS

The state space gets updated by the occurrence of any one of the following events: 1) customer arrival, 2) cancellation and 3) flight departure (t=0). Once the terminal state is reached, all actions will lead to the ending of the episode. Actions need to be taken only when a passenger arrives. The agent is said to be in a decision-making state at that instance. When a booking request is accepted, the seat for the corresponding fare class gets incremented by one. When it is denied, the seat for the corresponding fare class remains unchanged. In both cases, several cancellations in each fare class, following an uniform distribution, may have occurred since the last decision-making state. The corresponding number of seats must be deleted from the corresponding fare classes to get the updated state.

#### 3.3.4. REWARD FUNCTION AND DISCOUNT

The reward function gives back the fare associated with the passengers class if accepted or zero reward otherwise. Also, if a passenger has canceled since the last decision, then the fare of the passenger that cancelled will be subtracted

from the reward. At the time of departure, if there are more passengers booked than there is capacity on the flight, then the airline will have to bump some of the passengers in descending order of fare classes. Higher-class passengers are considered to be bumped first as they are typically not flying on a multi-leg itinerary. The cost of bumping a passenger from fare class $T$ is considered to be some multiple ($\beta_T$) of the passenger fare ($f_T$). This multiplication factor will be adjusted to test different cases. The number of passengers bumped from fare class $T$ ($\eta_T$) is computed at the end of each flight episode. The fares for each class have been set at $300 for the high class, $200 for the middle class, and $100 for the lowest class. These fares were set based on a flight from Chicago to New York, whose fares typically ranged from $100 - $300. The reward function ($R$) and the equation for the total bumping cost ($B$) are given below.

$$R = \begin{cases} f_T & \text{if } action = a_{+1}, \\ 0 & \text{if } action = a_{-1}, \\ -f_T & \text{at occurrence of cancellation}, \\ B & \text{at } t = 0. \end{cases} \qquad (6)$$

$$B = -\sum_{T=1}^{3} \eta_T \beta_T f_T \qquad (7)$$

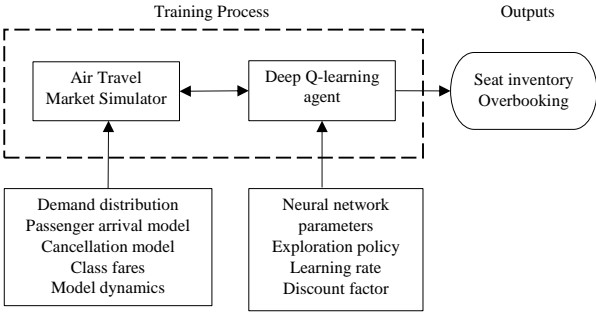

Figure 1. The components of our DRL powered RM system

## 4. Solution Method

In order to learn and evaluate a policy for seat inventory control and overbooking, a Deep RL (DRL) agent was trained and tested in an air travel market simulator as depicted in Fig. 1. The neural network is approximating the state-action value function needed for Q-learning. Based on the output of the neural network, the agent can decide which action to take. The DQN was implemented using the Keras (Chollet et al., 2015) and Keras-rl (Plappert, 2016) packages in Python. In Keras, a dense, feedforward neural network was created, consisting of an input layer, two hidden layers and an output layer. The input layer has six nodes, one node

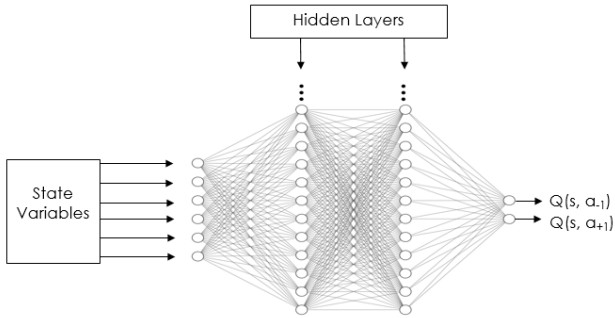

*Figure 2.* The deep neural network configuration used to approximate Q-values

for each variable in the state space and an additional bias node. The output layer has two linearly activated nodes, one node per action. Both of the hidden layers contained 128 ReLU-activated hidden neurons each. The structure of the neural network is depicted in Fig. 2. The performance of the agent was observed to depend strongly on the values of the neural network hyperparameters. After testing several settings of hyperparameters, the configuration described above was found to give the best performance.

In Keras-rl, a DRL agent was defined that updates the weights of the neural network after each interaction with the market simulator using Q-learning. A linear annealed $\varepsilon$-greedy policy was chosen as the exploration policy. In an $\varepsilon$-greedy policy, the agent chooses a random action with probability $\varepsilon$ or greedily chooses the currently estimated best action with probability $(1 - \varepsilon)$. In the linear annealed version of this policy, the value of $\varepsilon$ linearly decreases as the training progresses. For our purposes, the value of $\varepsilon$ was set to start at 1 and then linearly decrease to 0.1. So, the search policy started as choosing actions purely randomly and then ended choosing in a mostly greedy approach.

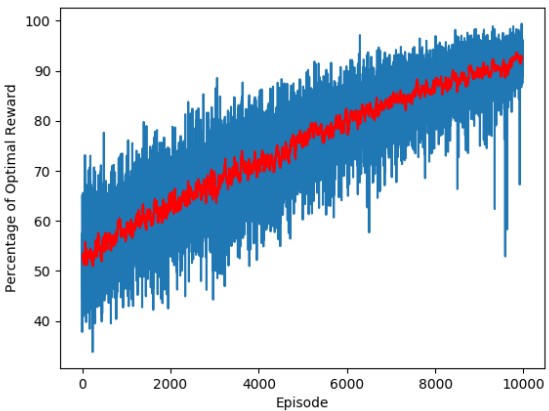

*Figure 3.* Percentage of optimal reward achieved during training

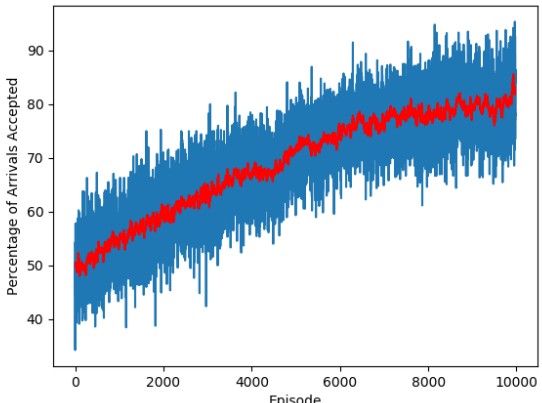

*Figure 4.* Percentage of arrivals accepted during training

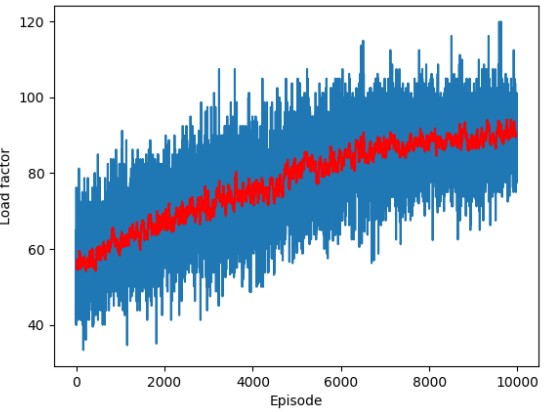

*Figure 5.* Load factor during training

## 5. Results

Several different test cases were simulated to evaluate the solution method. In all of these cases, the capacity of the flight is 80, the expected number of passenger booking requests is 100 and $\beta$ is 2 in each episode. This ensures the agent would generally have to deal with overbooking, such that the policy of accepting everyone will not be an optimal one. Also, this value of $\beta$ gave the agent sufficient negative incentive to not excessively overbook the flight. Three different fare class distributions of passengers were experimented, where the mean booking requests of the three fare classes are: [10, 30, 60], [60, 30, 10], [33, 33, 34]. For example, [10, 30, 60] means that on average 10 high class passengers, 30 middle class passengers, and 60 low class passengers will want to book. Again, each of these is modeled by a Poisson process, so each episode will vary in the actual number of passengers. Three different cancellation rates were tested: 0%, 10%, and

*Table 1.* Results for different test cases

| CANCELLATION RATE | CLASS DISTRIBUTION | AVG OPTIMAL REWARD (%) | AVG ACCEPTANCE RATE (%) | AVG LOAD FACTOR (%) |
|---|---|---|---|---|
| 0% | 1 | 97.628 | 77.670 | 96.365 |
| 0% | 2 | 93.651 | 81.715 | 101.738 |
| 0% | 3 | 97.046 | 77.434 | 95.092 |
| 10% | 1 | 95.564 | 88.014 | 97.995 |
| 10% | 2 | 93.661 | 89.418 | 100.608 |
| 10% | 3 | 94.855 | 78.707 | 88.787 |
| 20% | 1 | 96.080 | 96.090 | 95.984 |
| 20% | 2 | 89.868 | 81.949 | 81.909 |
| 20% | 3 | 94.637 | 93.407 | 93.317 |

20%.

For each of these cases, 10000 flight episodes of training data were generated. During training and testing, the performance of the agent was measured in terms of the percentage of theoretical optimal revenue it was able to obtain. Theoretically, the reward is maximal when the flight is just filled to capacity, starting with high fare class passengers and moving on to low fare ones, without any passengers getting bumped. The average acceptance rate expresses the average percentage of booking requests the agent accepted. Without any cancellations, the flight, on average, is just full when 80% of the booking requests are accepted. Similarly, with 10% and 20% cancellations rates, the average optimal acceptance rates are 88.88% and 100% respectively. The average load factor represents the average percentage to which flight was filled. When the load factor in any episode is greater than 100%, the flight is overbooked, resulting in excess passengers getting bumped and the agent receiving a corresponding negative reward.

A couple of interesting results can be seen from the table 1. First, despite not having any knowledge of the distribution of passenger arrivals and the cancellation rates of each fare class, the agent was still able to learn from experience and achieve near optimal results; in most cases, the agent is able to achieve over 93% of the optimal reward. The average acceptance rate is found be close to the average optimal one in most of the experiments, indicating that the agent was able to learn to overbook in such a way that the flight would be mostly full after cancellations. Ideally, the agent should try and achieve a 100% load factor to maximize revenue. It is evident that the agent is achieving close to this ideal in some cases. Note that in some cases with too many cancellations it might not be possible to fill up the flight completely.

Figures 3, 4, and 5 shows how the agent learns during the course of training for the case where the bumping cost factor is 2.0, the cancellation rate is 10%, and the fare class distribution is [33,33,34]. The red line in the plots represents

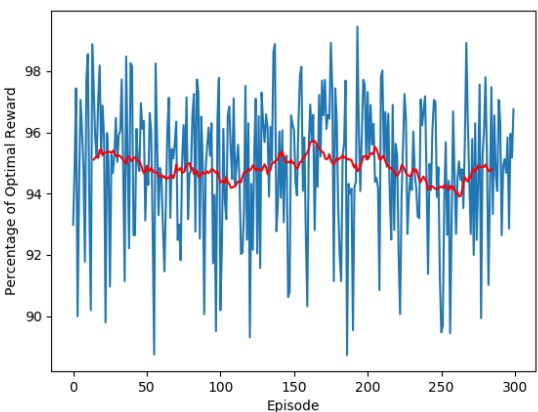

*Figure 6.* Percentage of optimal reward achieved during testing

the moving average. Predictably, at the start of training, the agent is unable to perform well, achieving only about 55% of optimal revenue by filling up the aircraft to about 60% and accepting around 50% of booking requests, as it is choosing its actions randomly and the neural network is initialized with random weights. However, as the training progresses, the agent starts learning from experience, gradually reducing exploration and increasing exploitation as reflected by the results increasing towards the optimal values. A variability of about 20% in the quantities was observed throughout the training. The agent can be seen to be converging to the optimal policy as the acceptance rate levels out around 83% and the load factor around 90% near the end of training.

The corresponding plots generated during the 300 episodes of testing phase in the same experiment are depicted in figures 6, 7, 8, and 9. The optimal revenue achieved is around 95%, with variations between 88% to 98% caused by the nature of the Poisson process that is being used to simulate the arrival of passengers. Still the reward stays above 90% of the optimal reward, despite this variability. It is interesting

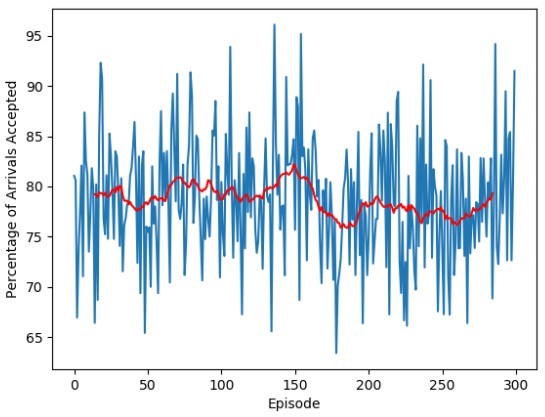

Figure 7. Percentage of arrivals accepted during testing

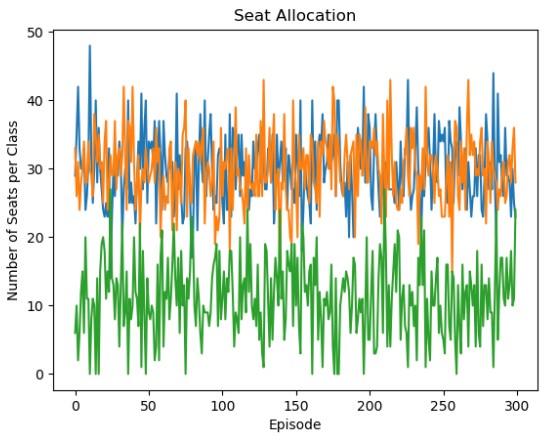

Figure 9. Seat Distribution of Fare Classes during testing

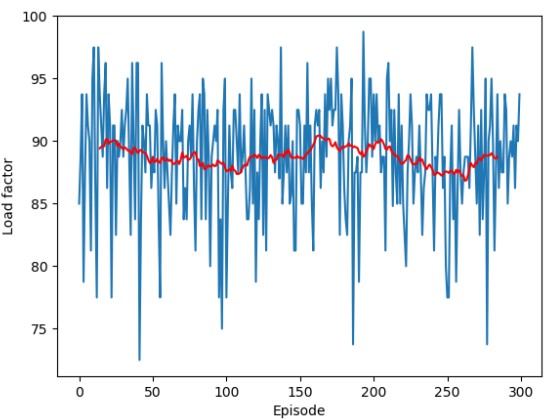

Figure 8. Load Factor during testing

to note in figure 9 how the agent learns to give priority to the high and middle fare class passengers above the low fare class ones. The agent accepts most of the booking requests from the high and middle fare class passengers, about 33 each, which reduces to 30 after cancellations on average, and books around 17 passengers from the low fare class, which closely resembles the optimal policy.

## 6. Conclusion

In this paper, a deep RL approach was used for airline RM in a single O-D market. The DRL agent achieved nearly optimal results in solving the airline seat inventory control and overbooking problem. A basic air travel market simulator was developed to model the demand distribution for multiple fare classes, concurrent arrival of passengers of each class with random cancellation in a given O-D market. A deep neural network was created to act as a global

approximator of the Q function, and Q-learning was used to train an agent to make decisions about accepting or denying passengers booking requests. The neural network was used to capture the nonlinearities of the Q-function in the large continuous state space. The agent was tested on numerous market scenarios. On average, the agent achieved higher than 93% of the theoretical optimal reward by overbooking properly so that the flight would be full after cancellations in most cases. The performance of the agent depended strongly on the accuracy of approximation of the Q-function, which, in turn, depended on the values of the neural network hyper-parameters. Additionally, the value of bumping cost factor, serving as an incentive for the agent to not excessively over-book, played a key role in the training of the agent. When it was low, the agent tended to accept most booking requests, resulting in high bumping cost and poor performance. This problem can however be addressed by larger exploration by the agent and lengthier training.

To embrace the full scope and range of aspects of the real-world airline RM problem, we are currently working towards training the agent on a network of interconnected O-D markets to tackle dynamic pricing along with seat inventory control and overbooking. The new set of actions allows the agent to vary the fares of the fare products with time till departure. Also, the action of denying booking requests has been replaced with withdrawing fare products for a given period of time. We are trying other RL algorithms for this airline RM problem. We expect the new methods will reproduce similar successes in this endeavor as were achieved in this paper. Ultimately, we firmly believe RL is a promising approach for real-world airline revenue management under uncertainty.

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
