# OpenReview forum: "Autonomous Airline Revenue Management: A Deep Reinforcement Learning Approach to Seat Inventory Control and Overbooking"
_ICML.cc/2019/Workshop/RL4RealLife — RL4RealLife 2019_

### Official Review · AnonReviewer1 · 2019-05-22
**Application of DQN to airline ticketing**

**Rating:** 3
**Confidence:** 4

**Review:**

This paper presents an application of DQN to the airline ticketing problem: deciding which ticket requests for different fare classes to accept to maximize revenue, given unknown future requests, cancellation rates, and penalties for overbooking. The paper presents the formulation of this problem as an MDP, a simulator representing the MDP, and evaluates DQN on a few variations of this task. In general, DQN comes close to the optimal performance on these tasks.

The paper cites many related works formulating this problem as an MDP or SMDP and applying value iteration or reinforcement learning. It's unclear how their formulation of the problem differs from these other related works. It's also unclear how their results compare to these approaches in terms of how close to optimal the performance is.

The empirical results also do not show any comparisons to related works or other algorithms.

Clarity:
The paper is very clear and easy to read.

Originality:
It's unclear how original the paper is. The MDP formulation looks to be very similar to Gasavi et al, but with the state missing exact knowledge of future arrival times. Clearly there's a lot of related work formulating this problem as an MDP, it would be good to know exactly how this formulation differs.

Significance:
If this work is much better than the related approaches and what is used in industry, then it could be quite significant. Again, this is unclear from the paper.

Pros:
- Real world application of DQN
- Clearly written paper

Cons:
- Not enough contrast and comparison with related work

---

### Decision · Program_Chairs · 2019-05-28

Accept